# High ADMA Is Associated with Worse Health Profile in Heart Failure Patients Hospitalized for Episodes of Acute Decompensation

**DOI:** 10.3390/medicina60050813

**Published:** 2024-05-15

**Authors:** Anamaria Vîlcea, Simona Maria Borta, Romana Olivia Popețiu, Rus Larisa Alexandra, Luminița Pilat, Dragoș Vasile Nica, Maria Pușchiță

**Affiliations:** 1Department of Internal Medicine, Faculty of Medicine, “Vasile Goldiș” Western University of Arad, Bulevardul Revoluției 94, 310025 Arad, Romania; simoborta@yahoo.com (S.M.B.); popetiur@gmail.com (R.O.P.); larisa_gal@yahoo.com (R.L.A.); mpuschita.mp@gmail.com (M.P.); 2Arad County Emergency Clinical Hospital, Str. Andrényi Károly Nr. 2-4, 310037 Arad, Romania; 3The National Institute of Research—Development for Machines and Installations Designed for Agriculture and Food Industry, Bulevardul Ion Ionescu de la Brad 6, 077190 București, Romania; nicadragos@gmail.com; 4Research Center for Pharmaco-Toxicological Evaluations, Faculty of Pharmacy, “Victor Babes” University of Medicine and Pharmacy, Eftimie Murgu Square No. 2, 300041 Timişoara, Romania

**Keywords:** acute decompensation episodes, chronic heart failure, asymmetric dimethylarginine, serum ADMA, length of hospitalization, renal function, cholesterol

## Abstract

*Background and Objectives*: episodes of acute decompensation in chronic heart failure (ADHF), a common health problem for the growing elderly population, pose a significant socio-economic burden on the public health systems. Limited knowledge is available on both the endothelial function in and the cardio-metabolic health profile of old adults hospitalized due to ADHF. This study aimed to investigate the connection between asymmetric dimethylarginine (ADMA)—a potent inhibitor of nitric oxide—and key health biomarkers in this category of high-risk patients. *Materials and Methods*: this pilot study included 83 individuals with a known ADHF history who were admitted to the ICU due to acute cardiac decompensation. Selected cardiovascular, metabolic, haemogram, renal, and liver parameters were measured at admission to the ICU. Key renal function indicators (serum creatinine, sodium, and potassium) were determined again at discharge. These parameters were compared between patients stratified by median ADMA (114 ng/mL). *Results*: high ADMA patients showed a significantly higher incidence of ischemic cardiomyopathy and longer length of hospital stay compared to those with low ADMA subjects. These individuals exhibited significantly higher urea at admission and creatinine at discharge, indicating poorer renal function. Moreover, their lipid profile was less favorable, with significantly elevated levels of total cholesterol and HDL. However, no significant inter-group differences were observed for the other parameters measured. *Conclusions*: the present findings disclose multidimensional, adverse ADMA-related changes in the health risk profile of patients with chronic heart failure hospitalized due to recurrent decompensation episodes.

## 1. Introduction

Heart failure is a chronic condition that develops when the weakened or stiffened heart muscle cannot pump blood throughout the body within the physiological pressure levels [1,2]. This clinical syndrome often leads to shortness of breath, fatigue, rapid weight gain, and swollen legs/abdomen due to fluid retention in the lungs and body tissues. The incidence of heart failure increases with age, being a leading cause of cardiac-related hospitalization in people aged 65 years and over [3] and accounting for a large part of the total expenditures allocated for treating cardiovascular diseases [1,4]. Unlike de novo acute heart failure, episodes of acute decompensation in chronic heart failure (ADHF) implies the sudden worsening of symptoms in individuals with a prior diagnosis of heart failure [5]. This condition needs prompt and comprehensive management to stabilize patients and prevent further complications [5]. Upon hospital admission, a thorough assessment and diagnostic tests are conducted to identify the underlying ADHF cause [2]. This can include laboratory tests such as electrolyte levels, renal function, and cardiac biomarkers, as well as imaging studies like chest X-ray and echocardiography [6]. Recurrent ADHF are associated with substantial economic burdens on individuals and healthcare systems [7], and this impact is likely to grow as the global population ages [8]. Studying ADHF within the context of an aging population is, therefore, critical for addressing the unique challenges, optimizing care delivery, improving outcomes for older adults with heart failure, and allocating resources efficiently to meet the needs of this vulnerable population [2,3,5].

The vascular endothelium plays a central role in cardiovascular health [9,10,11]. The flow-mediated dilatation (FMD) of the brachial artery is the gold standard for assessing systemic endothelial dysfunction; it is a direct, non-invasive, and simple method, with good sensitivity to early stages of endothelial dysfunction [12]. However, this technique is time-consuming, operator-dependent, and sensitive to various factors, such as blood pressure, heart rate, smoking, room temperature, and medications [13]. As a result, new biomarkers are needed to overcome these drawbacks. Ideally, these new markers should be more specific, accessible, and offer earlier detection or additional information to complement FMD.

Asymmetric dimethylarginine (ADMA), a potent inhibitor of nitric oxide production, is emerging as a potential major player in this arena [9]. It can be measured via a simple blood test, making it more accessible than FMD which requires ultrasound equipment and trained personnel [14]. In addition, ADMA offers improvement in specificity compared to FMD, since it provide scientists and medical personnel with more details about the potential causes of endothelial dysfunction [15]. This analogue of L-arginine is a risk factor for different cardiac pathologies, including hypertension, stroke, and coronary heart disease [16,17,18], and a pertinent predictor of mortality and major adverse events in such patients [19,20]. There is also evidence of its clinical relevance in individuals with ADHF. It has been found that these subjects often exhibit higher ADMA levels compared to healthy individuals [15,21]. Elevated ADMA is also associated with a worse prognosis in this category of cardiac patients during stay at an intensive care unit (ICU) [13,15]. However, the significance of ADMA in adults with cardiac diseases is far from being fully understood. Moreover, limited information exists about the link between ADMA and ADHF [22,23]. 

Cardiovascular parameters, such as blood pressure, heart rate, ejection fraction, and ventricular wall thickness play a crucial role in evaluating cardiac function and guiding treatment decisions for ICU patients [24]. Moreover, changes in markers like haemogram-derived indices, glucose levels, renal function indicators, lipid levels, and liver transaminases are associated with cardiac pathologies and can impact prognosis [25,26,27,28,29,30,31,32]. Nonetheless, information about the connection between these health markers and different degrees of endothelial dysfunction is currently lacking. We, therefore, aimed to contribute to a more refined understanding of the clinical relevance of ADMA in elderly persons with ADHF. Our hypothesis was that patients with different ADMA levels may display different blood, glycemic, renal, lipid, and hepatic profiles during ADHF. Investigating the link between ADMA and the aforementioned health profiles provides a more comprehensive picture of the health status of ADHF patients. This can lead to a better understanding of how ADMA influences the disease process and potentially guide treatment decisions or risk stratification.

## 2. Materials and Methods

### 2.1. Study Design

We conducted this exploratory, academic, single-site, pilot clinical study without industry involvement to understand differences in cardio-metabolic profiles of ICU-hospitalized, ADHF patients with different levels of serum ADMA. To the best of our knowledge, such basic information is not available in medical literature. This prospective observational study took place in the Department of Cardiology of the Arad County Clinical Hospital (Arad, Romania) and was run in agreement with the 1964 Declaration of Helsinki and later amendments. This study was approved by the Ethics Committees (IECs) of the two institutions involved, that is, the aforementioned hospital (approval No. 23/13 December 2018) and “Vasile Goldiș” Western University of Arad, Romania (approval No. 141/1 March 2019). The study population was selected from a cohort of elderly adults from the west part of Romania who were admitted to the ICU and had a medical history of heart failure with episodes of acute decompensation. The patients included in this study were selected from a larger pool of cardiac subjects who were hospitalized and treated in the Cardiac Intensive Care Unit of Arad County Clinical Hospital during the period of six months (from December 2019 to June 2020). All patients or their caregivers received and signed an informed consent form before any study-specific procedure commenced.

### 2.2. Patients and Measurements

Between December 2019 and June 2020, all consecutive cardiac patients with a medical history of ADHF who were admitted to the Cardiac ICU of Arad County Clinical Hospital due to acute episodes of cardiac decompensation were invited to participate in the present study. The main inclusion criteria were a medical history of chronic heart failure with acute decompensation episodes, a B-type natriuretic peptide concentration above 125 picograms per milliliter (pg/mL), and at least one clinical sign of volume overload (e.g., edema, pleural effusion, ascites). The main exclusion criteria were de novo heart failure, use of acetazolamide maintenance therapy or another proximal tubular diuretic, myocardial infarction less than one year prior to admission to hospitalization, valve disease requiring intervention, women who were pregnant, systolic blood pressure lower than 90 mm Hg, life expectancy <1 year due to non-cardiac pathology, and an estimated glomerular filtration rate (GFR) below 20 mL per minute per 1.73 m^2^ of body surface area.

Systolic blood pressure (SBP), diastolic blood pressure (DBP), and heart rate (HR) were determined at the time of admission to the Cardiac Intensive Care Unit of Arad County Clinical Hospital. Blood sampling was conducted as soon as possible after hospital admission (maximum 2 h post-admission). These samples were used to measure serum ADMA levels. Selected haemogram-derived indices included absolute leukocyte count (ALC), absolute neutrophile count (ANC), platelet-related parameters (RDW-CV, RDW-SD), and hemoglobin; key renal health markers included serum creatinine, GFR, serum urea, serum sodium, and serum potassium; lipid parameters included total cholesterol, LDL, HDL, and triglycerides; and liver markers included aspartate transaminase (AST) and alanine transaminase (ALT). The prompt collection of blood upon hospital admission was crucial for minimizing variability in ADMA levels that might otherwise occur with prolonged hospitalization and potential influence of external factors.

Ejection fraction (EF), left ventricular posterior wall thickness (LVPWT), and interventricular septum thickness (IVSd) were determined via echocardiography. Since kidney function is a significant factor in the management and prognosis of decompensated heart failure [33], blood samples were collected again at discharge and used to quantify serum creatinine, sodium, and potassium. Most biochemical analyses were performed at the Arad County Clinical Hospital, except for ADMA measurements which were done at the “Vasile Goldiș” Western University of Arad. All analyses were run in triplicates and only the mean values were taken into account. 

Blood samples were collected from each patient at admission and discharge, with volumes of 50 mL and 20 mL, respectively. For the ELISA process, 10 mL of the initial blood samples was incubated at 25 °C for 2 h in a test tube before being centrifuged at 1000× *g* for 15 min. The resulting serum was then stored at −20 °C until analysis. ADMA levels were measured using the Human ADMA ELISA Kit (Cusabio, Wuhan, China), which utilizes a sandwich ELISA approach and has a detection range of 7.8 ng/mL to 500 ng/mL. All assay procedures were performed as per the manufacturer’s instructions. The absorbances of standards and samples were measured at 450 nm using a M200 PRO multimode microplate reader (Tecan Group Ltd., Männedorf, Switzerland). For precise quantification of ADMA, a standard curve was constructed by plotting the log10-transformed concentrations against the corresponding optical densities of the positive controls. The calibration curves for the ELISA exhibited coefficients of determination (R^2^) greater than 0.96.

Sociodemographic data of the study population (sex, age, origin, smoking status) were also collected, as well as data on the presence of diabetes, ischemic cardiomyopathy, tachyarrythmia, pulmonary hypertension due to left heart disease (PH-LHD), and renal dysfunction. Excluding the patients with a known history of disease, new diabetes was diagnosed based on HbA1c greater than 6.5% and/or fasting plasma glucose above 125 mg/dL and/or random plasma sugar above 200 mg/dL [34]. Ischemic cardiomyopathy was defined as a reduced ability of the heart to pump blood throughout the body within the physiological pressure due to myocardial ischemia [35]. Tachyarrhythmia was defined as a heart rhythm with a ventricular rate of at least 100 beats/min [35]. PH-LHD was defined as known LHD or newly diagnosed LHD (as per national guidelines for LHD diagnosis/management) with a mean pulmonary artery pressure (mPAP) ≥ 25 mm and/or systolic pulmonary artery pressure (sPAP) ≥ 35 mm Hg and tricuspid regurgitant jet velocity >3.4 m/s [35]. Renal dysfunction was defined as a GFR below 60 mL per minute per 1.73 m^2^ of body surface area [35]. 

### 2.3. Statistical Analysis 

Statistical analysis was conducted with the Statistica version 8 software (StatSoft Inc., Tulsa, OK, USA), with a *p* value < 0.05 being considered significant. The study population was first divided into two strata using the median ADMA of the overall population as a cut-off point. This approach was used since there are no established cut-off points for an ADMA clinical significance [13] and, therefore, the median offers a data-driven way to categorize patients without needing an arbitrary value [36]. Chi2 (χ^2^) tests were used to assess inter-group differences in the distribution of subjects grouped based on sex, area of origin, current status of smoking, and presence of diabetes, ischemic cardiomyopathy, tachyarrythmia, PH-LHD, or renal dysfunction. Mann–Whitney U tests were then applied to cardiovascular parameters, haemogram-derived indices, glycemic indicators, renal markers, lipid parameters, and liver markers investigated [37]. 

## 3. Results

### 3.1. Characteristics of the Subjects

The study population comprised 44 males (53.01%) and 39 females (46.99%). Among the 83 patients enrolled in the study, eight were in NYHA functional class 1 (9.36%), 10 were in NYHA functional class 2 (12.04%), 29 were in NYHA functional class 3 (34.93%), and 36 were in NYHA functional class 4 (43.37%). These subjects were split into two groups using the median ADMA of the overall population (114 ng/mL) as a cut-off point: the low ADMA strata and the high ADMA strata included 42 patients and 41 patients, respectively. Their sociodemographic characteristics and health conditions (as categorical variables) are given in Table 1. No significant inter-group differences existed with respect to sex distribution, origin, smoking status, presence of diabetes, ischemic cardiomyopathy, tachyarrythmia, PH-LHD, and renal dysfunction (Table 1). Ischemic cardiomyopathy was almost four times more frequent in high ADMA subjects relative to low ADMA subjects, and this difference reached statistical significance (Table 1).

### 3.2. Variables Associated with High ADMA Levels

The median ADMA for the study population was 114 ng/mL, with this variable showing a large dispersion of the measured values (lower quartile: 71; upper quartile: 998). Median values (with lower and upper quartiles) for age, cardiovascular parameters (SBP, DBP, HR, EF, LVPWT, IVSd, NYHA class at admission, NYHA class at discharge), blood parameters (ALC, ANC, RDW-CV, RDW-SD, hemoglobin), glycemic control parameters (admission glycemia), renal function parameters (GFR, serum sodium at admission and discharge, serum potassium at admission and discharge), lipid parameters (LDL, triglycerides), and liver parameters (AST, ALT) are shown in Table 2. The reference range for these parameters are provided in the same table. The corresponding values for hospitalization time, serum urea, serum creatinine (both at admission and discharge) are shown in Figure 1a–c, and those for total serum cholesterol and HDL are shown in Figure 2a,b. By separately presenting data for non-significant (continuous) variables and significant (continuous) variables (see Table 1 and Figure 1 and Figure 2), we aimed to improve readability, optimize space, and effectively highlight the magnitude and direction of inter-strata differences for significant variables.

Most parameters shown in Table 2 were within the normal range, except for the median SBP and DBP which were above the upper normal limits. No significant inter-group differences existed with respect to age (Table 2). High ADMA subjects spent significantly more time in the hospital than subjects with low ADMA (Figure 1a; Mann–Whitney U test, 5 days (2; 10) vs. 4 days (1; 9), *p* = 0.042). The measured values for all cardiovascular parameters analyzed were similar between the two strata (Table 2), as well as the values for blood parameters investigated and for admission glycemia (Table 2).

Regarding the renal function, there were no significant inter-group differences in GFR, serum sodium (both at admission and discharge), and serum potassium (both at admission and discharge) (Table 2). However, high ADMA patients showed significantly elevated urea levels compared to low ADMA patients (Figure 1a; Mann–Whitney U test, 51.62 mg/dL (45.26; 66) vs. 40 mg/dL (32.40; 60), *p* = 0.042). Although the measured values for admission creatinine were similar between the two strata (Figure 1b; Mann–Whitney U test, 0.92 mg/dL (0.83; 1.18) vs. 1.01 mg/dL (0.87; 1.25), *p* = 0.098), at discharge, they were significantly heightened for the former category of subjects (Figure 1c; Mann–Whitney U test, 1.10 mg/dL (0.95; 1.35) vs. 0.90 mg/dL (0.75; 1.25), *p* = 0.040).

Statistical analysis of lipid parameters revealed no significant inter-strata differences in LDL and triglycerides (Table 2). Interestingly, total cholesterol was significantly higher in patients with elevated ADMA (Figure 2a; Mann–Whitney U test, 159.4 mg/dL (129; 2001.35) vs. 138.7 mg/dL (113; 175) *p* = 0.046). Similar results were obtained for HDL (Figure 2b; Mann–Whitney U test, 44 mg/dL (36; 54.2) vs. 38.5 mg/dL (32; 44), *p* = 0.023). However, liver-related parameters (AST, ALT) did not differ between the two groups.

## 4. Discussion

To the best of our knowledge, this is the first study to provide insights into connections between ADMA and key health metrics in cardiac patients hospitalized with ADHF. Previous research has linked elevated ADMA to worse outcomes in DCF patients [15,21]. However, these studies did not include cardiovascular, glycemic, renal, lipid, and hepatic biomarkers, although understanding their dynamics could help scientists clarify this interplay [38,39].

Most adults enrolled in this prospective observational study exhibited poor functional status, as indicated by the predominance of NYHA classes showing moderate or severe limitations. The number of males and females in the two ADMA strata was similar within the study population. The even sex ratio and focus on patients in poor condition strengthen the study’s internal validity and clinical relevance [40], allowing for a more generalizable understanding of how ADMA might relate to DCF, particularly in critically ill patients. High ADMA subjects showed significantly higher incidence of ischemic cardiomyopathy compared to low ADMA adults. One potential connection between ADMA and ischemic cardiomyopathy is the role of ADMA in promoting endothelial dysfunction, leading to vasoconstriction and reducing blood flow and oxygen delivery to the heart muscle [22]. However, this relationship is most probably multifactorial, involving a complex interplay between endothelial dysfunction, oxidative stress, inflammation, impaired NO signaling, fibrosis, remodeling, hemodynamic effects, and associated comorbidities [22,41].

Cardiac patients admitted to the ICU due to ADHF and exhibiting increased levels of serum ADMA experienced prolonged hospital stays compared to those with low ADMA concentrations. This finding is consistent with literature data. For example, Schulze et al. (2009) found that ADMA independently predicts the length of hospital stay in adults attending the emergency room. Of note, this metabolic byproduct revealed the highest predictive value for adults with cardiovascular conditions. The authors have also identified a significant direct correlation between ADMA and a composite endpoint consisting of complications/death during hospitalization [42]. In another study, Dückelmann et al. (2007) reported that patients with higher ADMA had a significantly higher risk of adverse cardiovascular events, including cardiac decompensation and mortality [43]. It is known that ADMA inhibits nitric oxide synthesis, thus promoting endothelial dysfunction, oxidative stress, inflammation, and increased cardiovascular disease risk [44]. Cardiac patients with high ADMA levels may hence display more severe endothelial dysfunction and underlying cardiovascular diseases, leading to more complex clinical presentations and longer hospital stays. On the other hand, these subjects may be more prone to developing complications during hospitalization, such as arrhythmias, myocardial infarction, stroke, or acute kidney injury [45,46]. These complications can prolong hospital stays, requiring additional monitoring, interventions, and treatments. 

Increased ADMA concentrations were accompanied by a less favorable renal panel compared to that seen for low serum ADMA. This profile was characterized by significantly elevated serum urea at ICU presentation and significantly increased creatinine at discharge versus the time of ICU admission. ADMA, a potent inhibitor of nitric oxide, can cause vasoconstriction, i.e., the narrowing of blood vessels, in the kidneys [46]. Reduced blood flow to the kidneys due to high ADMA in ADHF patients can potentially hinder their ability to filter waste products from the blood. In fact, older adults with ADHF can display low renal blood flow due to activation of the renin–angiotensin–aldosterone system (RAAS) [47]. This can reduce the blood flow, leading to decreased urea excretion, thus explaining the initial elevation of serum urea. 

On the other hand, it is possible that high ADMA may be a marker for pre-existing chronic kidney disease (CKD) [48]. CKD patients are known to already have reduced kidney function and decreased ability to excrete urea effectively [46]. During ADHF with high ADMA, the stress from the heart failure on the already compromised kidneys can further worsen urea excretion, yielding a more pronounced elevation of serum urea at presentation. In addition, this insult can cause a slower rise in creatinine initially (due to decreased production from lower muscle perfusion) and a more prominent rise upon recovery (as congestion and other factors improve). Moreover, our results may reflect the fact that urea, unlike creatinine, is not reabsorbed by the kidneys [48]. So, its rise is more immediate and reflects the immediate decline in kidney function due to elevated ADMA and heart failure. However, creatinine levels at discharge can be misleading, since kidney function might still be recovering and the true picture might emerge only with later follow-up tests [49]. 

Literature data support a less favorable post-discharge outcome for cardiac patients with elevated serum ADMA concentrations. Aronson et al. (2004) measured blood urea nitrogen (BUN), a surrogate for serum urea, and the BUN/creatinine (BUN/Cr) ratio in 541 DCI patients admitted to the hospital. It was found that the latter, but not the former marker, is related to post-discharge mortality [50]. Filippatos et al. (2007) revealed that baseline BUN, but not creatinine, is a powerful predictor of elevated mortality at 60 days post-discharge in ADHF adults [51]. Georgiopoulou et al. (2016) showed that high baseline serum creatinine and BUN are associated with death/cardiac (re)hospitalization risk [52]. By establishing, for the first time, a connection between high ADMA levels and renal dysfunction, our research contributes to a deeper understanding of the mechanisms underlying adverse outcomes in ADHF patients. These findings underscore the importance of monitoring ADMA levels and renal function in clinical management to potentially improve outcomes and reduce post-discharge risks in this category of high-risk patients.

The increase in total cholesterol and high-density lipoprotein (HDL) in ADHF subjects with elevated ADMA without any change in LDL or triglyceride levels is also noteworthy. ADMA is associated with inflammation [10], a common feature in ADHF [43]. Inflammation can increase HDL and total cholesterol levels while decreasing LDL clearance by the liver [53,54]. This could account for the higher total and HDL cholesterol in the high-ADMA group. It is also possible that the function of HDL particles in ADHF adults with elevated ADMA levels is impaired. This may explain why LDL levels were not found to be significantly lower despite increased HDL. Indeed, recent evidence supports that the quality of the HDL-c is a better marker of cardiovascular and death risk than the quantity of HDLs [55,56,57,58]. Taken together with the aforementioned findings, these data suggest that high serum ADMA concentrations in ADHF patients are associated with a higher likelihood of ischemic cardiomyopathy, prolonged hospital stays, poorer renal function, and less favorable lipid profile.

The main limitations of this study stem from its pilot, exploratory design. Firstly, the study employs a cross-sectional design with single time-point measurements for most variables. While this allows for the identification of associations, it does not establish a clear cause-and-effect relationship [59,60]. However, we have incorporated several longitudinal datasets focusing on key renal markers (see above). The analysis of these datasets suggests that the ADMA–ADHF interaction includes a renal component. However, it remains uncertain whether elevated ADMA levels are a cause or a consequence of ADHF [59]. Future research will utilize longitudinal designs to further explore the temporal variations in biomarkers and their relationships with ADMA levels, thereby providing a more comprehensive view of heart failure progression.

Secondly, the findings of this study are based on a moderate sample size of 83 individuals, which might limit their broader applicability. Generally, larger sample sizes are favored to increase statistical power and enhance the validity of new biomarkers, as they enable a more precise understanding of biomarker distribution [60]. However, in scenarios involving new biomarkers with substantial interquartile ranges, such as in our study, it is preferable to use smaller sample sizes of up to 45 per stratum. This approach is ideal for pilot studies that focus on exploration and feasibility rather than definitive outcomes, as their power calculations are not consistently reliable [61]. Moreover, smaller sample sizes, typically ranging from 10 to 40 participants per stratum, are frequently employed in similar exploratory pilot studies due to constraints related to time, budget, and participant burden [37,60,61]. It is also noteworthy that this study was conducted during the COVID-19 pandemic, which significantly challenged healthcare systems, complicating the recruitment of participants, delivery of treatments, and accurate data collection. 

Thirdly, this investigation has employed univariate, not multivariate, analysis, although the latter approach could elucidate the independent contribution of ADMA to health outcomes while controlling for potential confounders [61]. Besides resource constraints, univariate analysis was chosen to maintain simplicity and clarity in our approach. It allows for a straightforward interpretation of the data, focusing on clear, singular relationships without the entanglement of multiple variables. This method not only aligns with the exploratory nature of our study, but also ensures that any patterns or trends we identify are robust and more likely to be verifiable [37,60,61]. In fact, pilot exploratory studies typically avoid multivariate techniques mainly due to high risk of model overfitting (capturing noise, not patterns) and challenges related to interpreting complex results derived from sample sizes too small for reliable detection of true relationships and interactions between variables This preliminary phase sets a solid foundation for future research, where, with appropriate resources, larger sample sizes, and refined data collection methods, more comprehensive multivariate analyses can be conducted to expand upon our initial findings.

Fourthly, the single-site, observational design of this study might introduce biases originating from the specific demographics and clinical practices of our location, which could affect the robustness and generalizability of the present findings. Nevertheless, such a design is typical of pilot, exploratory studies due to its cost-effectiveness, which supports focused objectives, allows for stringent control over variables, and provides flexibility in execution [60]. Conducting the study within a single institution (location) also leverages local expertise and an in-depth knowledge of the patient population. The primary goal of exploratory studies like ours is to gather preliminary data, serving as a foundation for more comprehensive future research [61].

Lastly, as it is characteristic of pilot exploratory studies, our research aimed at hypothesis generation rather than comprehensive validation. The study was specifically designed to assess the potential of ADMA as a biomarker in ADHF outcomes. As a result, we did not incorporate additional clinical data such as medication history, comorbidities, and patient outcomes post-discharge, nor did we include other biomarkers like those for inflammation or endothelial dysfunction. While these factors are undoubtedly significant, their inclusion would have expanded the study’s scope excessively, potentially overshadowing the targeted investigation of ADMA.

Despite its limitations, this clinical investigation has several important strengths. Apart from being among the few studies linking ADMA to patients with ADHF, a key strength is the holistic, multidimensional approach used to understand the health status of these patients. ADHF is influenced by a complex array of physiological factors, not just one; thus, our use of multiple biomarkers provides a fuller picture of patient health and contributing factors. Another strength is that our work focuses on a specific population of interest without confounding factors, e.g., recent myocardial infarctions (MI). For example, exclusion of patients with recent MI allowed us to avoid the potential confounding effects related to the administration of drugs that can interfere with the parameters analyzed. Such an example is the dual antiplatelet therapy (DAPT), combining aspirin with a P2Y12 inhibitor, which can affect metabolic and renal parameters [35]. Moreover, we incorporated longitudinal measurements for renal markers highly relevant in a cardiac context (serum creatinine, sodium, and potassium). This is important, since the kidneys are highly responsive to changes in cardiac dynamics and are among the most affected organs in ADHF [33,46]. This combination of cross-sectional and longitudinal data collection enhances our ability to observe the impact of ADHF on renal function over time, enhancing our understanding of the progression and management of this heart condition. By integrating renal markers with other cardiac and metabolic indicators, our study presents a thorough overview of the the interdependencies within physiological systems affected by ADHF. This integrated approach may help identify potential therapeutic targets and improve patient management strategies.

In conclusion, future research should aim to build on the foundational insights gained from this study by expanding its scope to include multisite investigations. Such expansions would enhance the diversity of patient demographics and clinical practices examined, providing a more comprehensive understanding of ADMA’s role in ADHF across different populations. Longitudinal studies extending beyond the initial hospitalization period could also provide valuable data on long-term patient outcomes and the efficacy of various treatment protocols. Through these endeavors, the goal will be not only to confirm the findings of the current study, but also to explore innovative therapeutic strategies, ultimately contributing to improved clinical guidelines and patient care in heart failure management. Moreover, integrating a broader range of clinical data is another crucial next step. Expanding these datasets to include variables such as medication history, comorbidities, and patient outcomes post-discharge will allow us to more deeply analyze the interactions of ADMA with these factors. This will enable medical scientists and clinicians to understand the nuances of how ADMA influences the management and prognosis of ADHF under varied clinical conditions. Additionally, this approach will provide pertinent, realistic data for predictive modeling, potentially equipping clinicians with enhanced tools to more accurately predict ADHF outcomes.

## 5. Conclusions

This study marks a significant step in understanding the intricate relationship between ADMA and health metrics in cardiac patients hospitalized due to recurrent ADHF. It integrates cardiovascular, metabolic, haemogram, renal, and liver biomarkers to offer new insights into this complex condition. Our findings corroborate previous research linking elevated ADMA levels with worse outcomes in heart failure patients. Notably, higher ADMA levels were associated with increased ischemic cardiomyopathy and longer hospital stays, highlighting its importance in prognosis. Elevated ADMA levels correlated with poorer renal function, as evidenced by higher serum urea and creatinine levels at ICU admission and discharge, respectively. This suggests that ADMA’s inhibition of nitric oxide synthesis contributes to renal dysfunction in ADHF by reducing renal blood flow and filtration capacity. Changes in lipid metabolism were also seen, with increases in total cholesterol and high-density lipoprotein (HDL) in patients with high ADMA levels. These alterations may reflect an inflammatory response in ADHF, impacting disease progression and outcomes. These findings underline the necessity for further research to confirm these relationships and explore potential therapeutic targets for heart failure management.

## Figures and Tables

**Figure 1 medicina-60-00813-f001:**
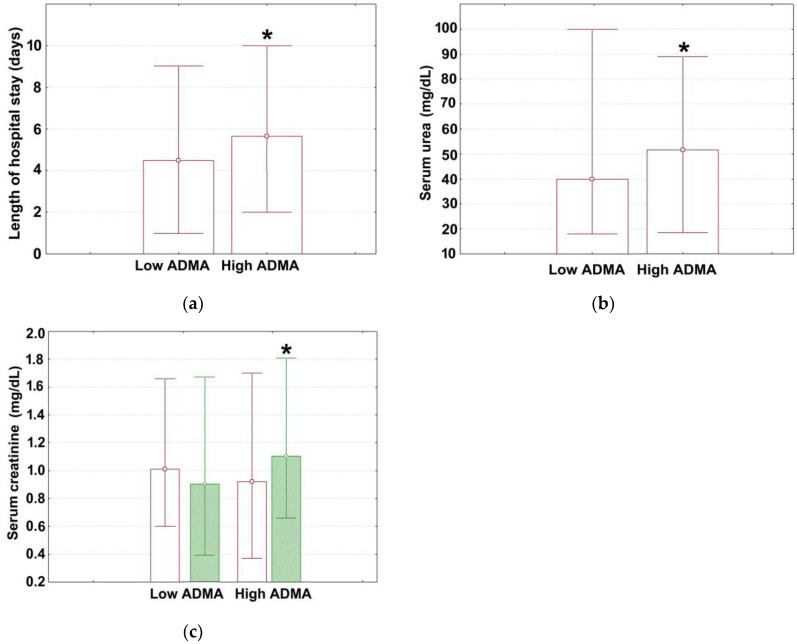
The measured values for (**a**) length of hospital stay, (**b**) serum urea, and (**c**) serum creatinine in different ADMA strata. Data are given as median (box) with lower and upper quartiles (error bars). Marked boxes (*) in (**a**,**b**) denote significant differences compared to low ADMA adults (Mann–Whitney U test, ***—*p* < 0.001, **—*p* < 0.01, *—*p* < 0.05). In (**c**), creatinine levels were measured both at admission (brown box) and discharge (green tint). Marked boxes (*) in subfigure (**c**) indicate significant differences at discharge compared to the time of admission (Mann-Whitney U test, ***—*p* < 0.001, **—*p* < 0.01, *—*p* < 0.05).

**Figure 2 medicina-60-00813-f002:**
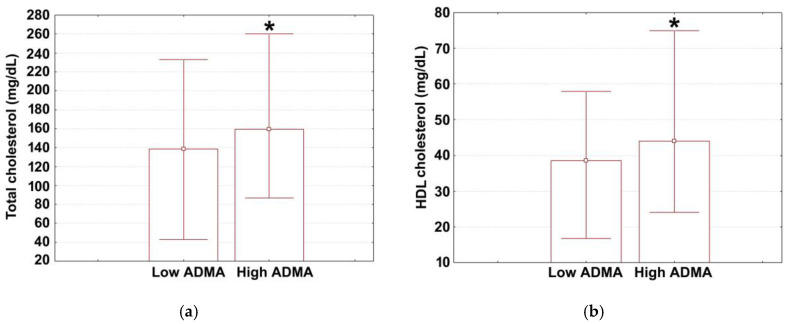
The measured values for (**a**) total cholesterol and (**b**) HDL cholesterol levels in different ADMA strata. Data are given as median (box) with lower and upper quartiles (error bars). Marked boxes (*) denote significant differences compared to low fibrinogen adults (Mann–Whitney U test, ***—*p* < 0.001, **—*p* < 0.01, *—*p* < 0.05).

**Table 1 medicina-60-00813-t001:** Sociodemographic characteristics and health conditions in different ADMA strata.

Characteristic	Strata	Low ADMA Patients	High ADMA Patients	*p*
Sex	Male	32 (76.91%)	34 (82.92%)	0.588
	Female	10 (23.81%)	7 (17.08%)	
Origin	Rural	21 (50%)	27 (65.86%)	0.125
	Urban	21 (50%)	23 (56.09%)	
Smoking status	Yes	24 (57.41%)	28 (68.29%)	0.365
	No	18 (42.86%)	13 (31.71%)	
Diabetes	Yes	16 (38.09%)	16 (39.02%)	1
	No	26 (61.91%)	25 (60.98%)	
Ischemic cardiomyopathy	Yes	11 (26.19%)	37 (90.24%)	0.047 *
	No	31 (73.81%)	4 (9.76%)	
Tachyarrythmia	Yes	13 (30.95%)	20 (48.78%)	0.144
	No	29 (69.05%)	21 (50.22%)	
PH-LHD	Yes	22 (52.39%)	19 (46.34%)	0.662
	No	20 (47.61%)	22 (53.66%)	
Renal dysfunction	Yes	22 (52.39%)	20 (48.78%)	0.624
	No	20 (47.61%)	21 (50.22%)	

PH-LHD, pulmonary hypertension due to left heart disease. Data are shown as absolute values with the corresponding percentage in parentheses. Marked values (*) show significant differences compared to low ADMA patients (χ^2^ tests, ***—*p* < 0.001, **—*p* < 0.01, *—*p* < 0.05).

**Table 2 medicina-60-00813-t002:** Measured values for selected parameters in different ADMA strata.

Characteristic	Low ADMA Patients	High ADMA Patients	Reference Range	*p*
Age	69.5 (64; 79)	70 (60; 77)		0.396
SBP (mm Hg)	142.5 (130; 155)	140 (120; 160)	90–130	0.761
DBP (mm Hg)	80 (73; 90)	82 (120; 160)	60–80	0.577
HR (beats/min)	91 (80; 120)	90 (80; 108)	60–100	0.741
EF (%)	40 (26; 55)	45 (35; 54)	>50	0.471
IVSd (cm)	1.2 (1; 1.3)	1.1. (1; 1.2)	0.6–1.2	0.407
LVPWT (cm)	1.15 (1.1.; 1.3)	1.1 (1; 1.2)	0.8–1.1	0.155
ALC (10^3^ cells/μL)	1.315 (1.080; 2.150)	1.660 (1.130; 2.210)	1–4	0.454
ANC (10^3^ cells/μL)	5.375 (3.710; 6.850)	6.470 (4.870; 7.940)	2–8	0.104
RDW-CV (%)	15.1 (14.3; 16.4)	14.55 (13.65; 16.20)	11.5–15.4	0.116
RDW-SD (fL)	46.4 (44.6; 51))	45.6 (42.5; 49.5)	39–46	0.229
Hemoglobin (g/dL)	13.25 (12.2; 14)	13 (11.8; 14)	12–18	0.362
Glycemia (mg/dL)	138 (113.8; 162.7)	126 (107; 155)	<200	0.229
GFR (mL/min)	70.65 (47.5; 93.3)	72.85 (61.5; 91.95)	>90	0.098
Serum Na, admission (mmol/L)	140.5 (138; 142)	137 (134; 141)	135–145	0.189
Serum Na, discharge (mmol/L)	139 (136; 142)	137 (134; 141)	135–145	0.110
Serum K, admission (mmol/L)	4.2 (4; 4.6)	4.3 (3.9; 4.7)	3.5–5	0.775
Serum K, discharge (mmol/L)	4.1 (3.9; 4.4)	4.3. (3.9; 4.6)	3.5–5	0.137
LDL (mg/dL)	94 (71; 121)	96 (75.5; 123.40)	<100	0.755
Triglycerides (mg/dL)	103.5 (80; 135)	105 (90; 155)	<150	0.676
AST (UI/L)	29 (20; 44)	28 (21; 40)	5–56	0.775
ALT (UI/L)	21.1 (16.4; 36)	26 (16; 50)	9–40	0.609

Data are given as median values with lower and upper quartiles in parentheses. Marked values (*) indicate significant differences compared to low ADMA patients (Mann–Whitney U tests, ***—*p* < 0.001, **—*p* < 0.01, *—*p* < 0.05).

## Data Availability

All the data generated or analysed during this study are included in this published article.

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
