# Peer review of "High ADMA Is Associated with Worse Health Profile in Heart Failure Patients Hospitalized for Episodes of Acute Decompensation"

_medicina, 2024, doi:10.3390/medicina60050813_

Round 1

Reviewer 1 Report

Comments and Suggestions for Authors

In this manuscript Vilcea et al evaluated the association between ADMA and a series of health parameters in patients hospitalized for acute heart failure. They stratified patients (83 individuals) based on the median value of ADMA of their small population. In patients with high ADMA levels, worse renal function and worse lipid profile were found. The paper definitely requires improvement in many aspects. Here’s a list of issues to be addressed before publication:

1)      The introduction section is very long and fails in summarizing the main aspects of the subject addressed by the article; I would suggest to rewrite it completely in a much more targeted way. Info regarding HF epidemiology, general physiology of the vascular endothelium and obvious knowledge regarding cardiovascular parameters should be removed. In addition, data regarding the population included in the study should be reported in the methods and results section, as appropriate.

1)      It is not appropriate to consider this study a clinical trial as this is a prospective observational study. Please correct.

3)      I would suggest to include de novo HF among exclusion criteria and chronic heart failure among inclusion criteria. Please note that decompensated cardiac failure is not an appropriate definition of patients with history of HF. I would suggest the authors to improve their word choice (e.g. as suggested by the most recent definition of HF) and to see below the comment regarding appropriate terminology.

4)      The statistical analysis paragraph provides redundant information (e.g. not useful to provide all variables that were checked for differences between groups, please summarize them) whereas no info regarding the definition of some variables is provided (e.g. pulmonary hypertension, cardiomyopathy).

5)      10% of patients were in NYHA class I, a strange finding in a population hospitalized for AHF. Please comment.

6)      It is not clear what the authors mean when they refer to “Cardiomyopathy”. Is it ischemic cardiomyopathy, dilated cardiomyopathy or something else? Please note that this is not an appropriate terminology in the field of HF. The same applies to pulmonary hypertension, as this is not appropriately defined.

7)      When talking about results related to biomarkers, please report results in their extended form in text or tables as it is not clear from the figures whether the presented data are reliable.

8)      I would suggest the authors to  not come to conclusions that are not supported by the findings of the present paper. No role for changes in medications or different therapeutic approaches should be established based on ADMA levels.

Comments on the Quality of English Language

English language requires some revisions; here some examples

1)      Lipid profile instead of lipidic profile.

2)      Please note that TAS and TAD are not appropriate abbreviation con blood pressure; the same applies to TGO, TGP, ISV and many others reported

3)      Line 160 not clear what DCI means

4)      Appropriate terminology should be used: e.g. Heart failure instead of decompensated cardiac failure; please also correct abbreviations accordingly.

5)      Line 180 “When then applied” rather than “were next applied”

Author Response

First we want to thank the Reviewer for his/her thorough review of our manuscript and for the excellent suggestions that we received. We have made a concerted effort to adequately respond to each suggestion received from the reviewer in such a short time. We also want to mention that taking into the reviewer’s comments we have changed the title of the manuscript to better reflect the present medical terminology related to patients with a history of heart failure and recurrent episodes of acute decompensation: “High ADMA is Associated with Worse Health Profile in Heart Failure Patients Hospitalized for Episodes of Acute Decompensation”. We do believe that present title better reflects the topic of the paper and should meet the reviewer’s requests.

Q1. The introduction section is very long and fails in summarizing the main aspects of the subject addressed by the article; I would suggest to rewrite it completely in a much more targeted way. Info regarding HF epidemiology, general physiology of the vascular endothelium and obvious knowledge regarding cardiovascular parameters should be removed. In addition, data regarding the population included in the study should be reported in the methods and results section, as appropriate.

R1. We have improved this part of the initial manuscript as suggested by the reviewer. More precisely, the 1st and the 2nd paragraphs from the initial manuscript were merged together, and the info regarding HF epidemiology was excluded; the 3 paragraph from the initial manuscript was shortened accordingly; and the 5th and the 6th paragraphs from the initial form of the manuscript were also merged, with the excess text being removed (relocated):

” Heart failure is a chronic condition that develops when the weakened or stiffened heart muscle cannot pump blood throughout the body within the physiological pressure levels [1]. This clinical syndrome often leads to shortness of breath, fatigue, rapid weight gain, and swollen legs/abdomen due to fluid retention in the lungs and body tissues. The incidence of heart failure increases with age, being a leading cause of cardiac-related hospitalization in people aged 65 years and over [3]; and accounting for a large part of the total expenditures allocated for treating cardiovascular diseases [1,4]. Unlike de novo acute heart failure, episodes of acute decompensation in chronic heart failure (ADHF) implies the sudden worsening of symptoms in individuals with a prior diagnosis of heart failure [5]. This condition needs prompt and comprehensive management to stabilize patients and prevent further complications [5]. Upon hospital admission, a thorough assessment and diagnostic tests are conducted to identify the underlying ADHF cause [2]. This can include laboratory tests such as electrolyte levels, renal function, and cardiac biomarkers, as well as imaging studies like chest X-ray and echocardiography [6]. Recurrent ADHF are associated with substantial economic burdens on individuals and healthcare systems [7], and this impact is likely to grow as the global population ages [8]. Studying ADHF within the context of an aging population is therefore critical for addressing the unique challenges, optimizing care delivery, improving outcomes for older adults with heart failure, and allocating resources efficiently to meet the needs of this vulnerable population [2,3,5].

The vascular endothelium plays a central role for cardiovascular health [9–11]. The flow-mediated dilatation (FMD) of the brachial artery is the gold standard for assessing systemic endothelial dysfunction; it is a direct, non-invasive, and simple method, with good sensitivity to early stages of endothelial dysfunction [12]. However, this technique is time-consuming, operator dependent, and sensitive to various factors, such as blood pressure, heart rate, smoking, room temperature, and medications [13]. As a result, new biomarkers are needed to overcome these drawbacks. Ideally, these new markers should be more specific, accessible, and offer earlier detection or additional information to complement FMD.

Asymmetric dimethylarginine (ADMA), a potent inhibitor of nitric oxide production, is emerging as a potential major player in this arena [9]. It can be measured via a simple blood test, making it more accessible than FMD which requires ultrasound equipment and trained personnel [14]. In addition, ADMA offers improvement in specificity compared to FMD since it provide scientists and medical personnel with more details about the potential causes of endothelial dysfunction [15]. This analogue of L- arginine is a risk factor for different cardiac pathologies, including hypertension, stroke, and coronary heart disease [16–18], and a pertinent predictor of mortality and major adverse events in such patients [19,20]. There is also evidence for its clinical relevance for heart failure. It was thus found that these subjects often exhibit higher ADMA levels compared to healthy individuals [15,21]. Elevated ADMA is also associated with worse prognosis in ADHF patients during stay at an intensive care unit (ICU) [13,15]. However, the significance of ADMA in adults with cardiac diseases is far from being fully understood. Moreover, limited information exists about the link between ADMA and ADHF [22,23].

Cardiovascular parameters, such as blood pressure, heart rate, ejection fraction, and ventricular wall thickness play a crucial role in evaluating cardiac function and guiding treatment decisions for ICU patients [24]. Moreover, changes in markers like haemogram-derived indices, glucose levels, renal function indicators, lipid levels, and liver transaminases are associated with cardiac pathologies and can impact prognosis [25–32]. Nonetheless information about the connection between these health markers and different degrees of endothelial dysfunction is currently lacking. We therefore aimed to contribute to a more refined understanding of the clinical relevance of ADMA in elderly persons with ADHF. Our hypothesis was that patients with different ADMA levels may display different blood, glycemic, renal, lipid, and hepatic profile during ADHF. Investigating the link between ADMA and the aforementioned health profiles provides a more comprehensive picture of the health status of ADHF patients. This can lead to a better understanding of how ADMA influences the disease process and potentially guide treatment decisions or risk stratification.”

Q2: It is not appropriate to consider this study a clinical trial as this is a prospective observational study. Please correct.

R2: The reviewer is correct, and we have made the necessary corrections in the revised form of the manuscript. Please, see e.g. the section Abstract, p. 1.

Q3: I would suggest to include de novo HF among exclusion criteria and chronic heart failure among inclusion criteria. Please note that decompensated cardiac failure is not an appropriate definition of patients with history of HF. I would suggest the authors to improve their word choice (e.g. as suggested by the most recent definition of HF) and to see below the comment regarding appropriate terminology.

R3: We have improved this part of the initial manuscript and included exclusion and inclusion criteria in the manuscript as requested by the reviewer – see the section Materials and Methods, subsection 2.2., 1st paragraph:

”… The main inclusion criteria were a medical history of chronic heart failure with acute decompensation episodes, a B-type natriuretic peptide concentration above 125 picograms per milliliter (pg/mL) and at least one clinical sign of volume overload (e.g., edema, pleural effusion, ascites). The main exclusion criteria were de novo heart failure, …”

In addition, throughout the manuscript the term ”decompensated cardiac failure” (or DCI) was changed to ”heart failure”; and the term ”acutely decompensated chronic heart failure” was replaced with ” episodes of acute decompensation in chronic heart failure” or similar terms.

Q4: The statistical analysis paragraph provides redundant information (e.g. not useful to provide all variables that were checked for differences between groups, please summarize them) whereas no info regarding the definition of some variables is provided (e.g., pulmonary hypertension, cardiomyopathy).

R4: We understand the reviewer’s point of view, and have made the appropriate changes in the revised form of the manuscript. First, the information redundant information was removed– see the section Materials and Methods, subsection 2.3. – Statisticla analysis, 1st paragraph:

”… Chi2 (χ2) tests were used to assess inter-group differences in the distribution of subjects grouped based on sex, area of origin, current status of smoking, and presence of diabetes, ischemic cardiomyopathy, tachyarrythmia, PH-LHD, or renal dysfunction. Mann-Whitney U tests were then applied for cardiovascular parameters, haemogram-derived indices, glycemic indicators, renal markers, lipid parameters, and liver markers investigated [37] ”.

Moreover, a new paragraph was introduced at subsection 2.2. - Patients and Measurements as the last (and the 3rd) paragraph. This paragraphs introduces the categorical variables used in this study; as well as their definition (for cardiac variables those used according to the national guidelines for cardiac diagnosis).

”… Sociodemographic data of study population (sex, age, origin, smoking status) were also collected, as well as data on the presence of diabetes, ischemic cardiomyopathy, tachyarrythmia, pulmonary hypertension due to left heart disease (PH-LHD), and renal dysfunction. Excluding the patients with a known history of disease, new diabetes was diagnosed based on HbA1c greater than 6.5%, and/or fasting plasma glucose above 125 mg/dL, and/or random plasma sugar exceeding 200 mg/dL [34]. Ischemic cardiomyopathy was defined as reduced ability of heart to pump blood throughout the body within the physiological pressure due to myocardial ischemia [35]. Tachyarrhythmia was defined as a heart rhythm with a ventricular rate of at least 100 beats/min [35]. New PH-LHD was defined as a mean pulmonary artery pressure (mPAP) ≥ 25 mm, and/or systolic pulmonary artery pressure (sPAP) ≥ 35 mm Hg; and tricuspid regurgitant jet velocity > 3.4 m/s [35]. Renal dysfunction was defined as a GFR below 60 mL per minute per 1.73 m2 of body-surface area [35].”

Q5: 10% of patients were in NYHA class I, a strange finding in a population hospitalized for AHF. Please comment.

R5: The results reflect the status of patients at admission into ICU, as recorded by clinicians. We believe that a potential explanation for this finding is that these subjects displayed valvulopathies besides the main inclusion criteria, that is, a known history of chronic heart failure and acute decompensation episodes, elevated B-type natriuretic peptide concentration above 125 picograms per milliliter (pg/mL), and clinical signs of volume overload (e.g., edema, pleural effusion, ascites). However, we did not verify if this hypothesis is correct or not, since this it was outside the pupose of our paper.

Q6: It is not clear what the authors mean when they refer to “Cardiomyopathy”. Is it ischemic cardiomyopathy, dilated cardiomyopathy or something else? Please note that this is not an appropriate terminology in the field of HF. The same applies to pulmonary hypertension, as this is not appropriately defined.

R6: Agree. We refered to ischemic cardiomyopathy. We have, accordingly, incorporated the requested details in the revised form of the manuscript. Thus, the term cardiomyopathy was replaced with ischemic cardiomyopathy; this term was also defined at subsection 2.2. - Patients and Measurements as the last (and the 3rd) paragraph – see response to question Q4 (above). The same applies to to pulmonary hypertension; the term was changed to pulmonary hypertension due to left heart disease and an appropriate definition was included - see the same part of the revised manuscript; and the esponse to question Q4 (above).

Q7: When talking about results related to biomarkers, please report results in their extended form in text or tables as it is not clear from the figures whether the presented data are reliable.

R7: We have chosen to present in Table 1 data for variables that showed no significant differences between patients with low ADMA and high ADMA. The results for variables showing signficant inter-group differences are given in Figures 1 and 2. To address the requests of the reviewer we reported the results for the latter (significant) variables so that they also included the median values and the interquartile range for both strata (using the same format as in Table 1). Please section Results, subsection 3.2., last three paragraphs. e.g.,

”However, high ADMA patients showed significantly elevated urea levels compared to low ADMA patients (Figure 1A; Mann-Whitney U test, 51.62 mg/dL (45.26; 66) vs. 40 mg/dL (32.40; 60), p = 0.042). Although the measured values for admission creatinine were similar between the two strata (Figure 1B; Mann-Whitney U test, 0.92 mg/dL (0.83; 1.18) vs. 1.01 mg/dL (0.87; 1.25), p = 0.098), at discharge, they were significantly heightened for the former category of subjects (Figure 1C; Mann-Whitney U test, 1.10 mg/dL (0.95; 1.35) vs. 0.90 mg/dL (0.75; 1.25), p = 0.040).”

Q8: I would suggest the authors to not come to conclusions that are not supported by the findings of the present paper. No role for changes in medications or different therapeutic approaches should be established based on ADMA levels.

R8: We understand the reviewer’s point of view, and have made the necessary corrections in the revised form of the manuscript - see section Discussion, the last paragraph. More precisely, the last two paragraphs from the initial version of the manuscript were merged together and the text related to changes in medications or different therapeutic approaches (lines 417-439) was removed.

Q9: Comments on the Quality of English Language

English language requires some revisions; here some examples

1) Lipid profile instead of lipidic profile.

 - We have, accordingly, done the requested changes in the revised version pf the manuscris.

2) Please note that TAS and TAD are not appropriate abbreviation con blood pressure; the same applies to TGO, TGP, ISV and many others reported

 - The appropriate changes have been performed in the revised version of the manuscript, with the aforemnetioned acronyms being replaced by those commonly used in medical literature; that is, TAS was replaced with SBP, TAD with DBP, TGO with AST, TGP with ALT, etc.

3) Line 160 not clear what DCI means

 - the term DCI was changed to heart failure – as suggested by the reviewer below and above ; e.g., Q3 (see the revised version of the manuscript). In addition, throughout the manuscript the term „acutely decompensated chronic heart failure” was replaced with „episodes of acute decompensation in chronic heart failure”

4) Appropriate terminology should be used: e.g. Heart failure instead of decompensated cardiac failure; please also correct abbreviations accordingly.

- OK. The appropriate changes were made in the revised version of the manuscript.

5) Line 180 “When then applied” rather than “were next applied”

 - We thank the reviewer for pointing this out. The reviewer is correct, and we have made the necessary corrections in the revised form of the manuscript

In addition, we have rewritten a large part of the initial manuscript , improved grammar and English to improve the readability of text. We firmly believe that the Reviewer’s comments and suggestions have significantly improved this manuscript. We hope that the Reviewer find the revised version of the manuscript suitable for publication in Medicina.

Anamaria Vîlcea

Department of Internal Medicine, Faculty of Medicine, ‘Vasile Goldiș’ Western University of Arad, Romania, Bulevardul Revoluției 94, Arad 310025, Romania.

Arad County Emergency Clinical Hospital, Romania, Str. Andrényi Károly Nr. 2‑4, 310037 Arad, Romania

Reviewer 2 Report

Comments and Suggestions for Authors

Authors of this interesting study  investigated the clinical relevance of ADMA in elderly patients with ADCHF. Manuscript is generally well written, but I have few comments:

  1. Introduction section can be shortened, especially the part from line 93 -line 110. 

  2. Why did the authors choose to exclude patients with myocardial infarction less than 1 year prior index hospital admission?

  3. Was there any difference in the outcome between patients with high and low ADMA levels?

  4. Will authors suggest that ADMA can be used in the ICU as a prognostic biomarker?

Methodology is well written, as well as discussion section and conclusion. results are clearly presented. references are adequate.

Thank you

Author Response

First, we want to thank the Reviewer for his/her thorough review of our manuscript and for the excellent suggestions that we received. Please find below our response letter explaining, point-by-point, the changes made in response to the critiques/suggestions that we received. We have made a concerted effort to adequately respond to each suggestion received from the reviewer in such a short time.

Q10: Introduction section can be shortened, especially the part from line 93 - line 110.

R10. As suggested by the reviewer, we have shortened the text at lines 93-110. Please, see the section Introduction – the 5th and the 6th paragraphs from the initial form of the manuscript were merged, with the excess text being removed. In fact, the entire Introduction section was rewritten (taking also into account the suggestions of the 1st reviewer).

”Cardiovascular parameters, such as blood pressure, heart rate, ejection fraction, and ventricular wall thickness play a crucial role in evaluating cardiac function and guiding treatment decisions for ICU patients [24]. Moreover, changes in markers like haemogram-derived indices, glucose levels, renal function indicators, lipid levels, and liver transaminases are associated with cardiac pathologies and can impact prognosis [25–32]. Nonetheless information about the connection between these health markers and different degrees of endothelial dysfunction is currently lacking. We therefore aimed to contribute to a more refined understanding of the clinical relevance of ADMA in elderly persons with ADHF. Our hypothesis was that patients with different ADMA levels may display different blood, glycemic, renal, lipid, and hepatic profile during ADHF. Investigating the link between ADMA and the aforementioned health profiles provides a more comprehensive picture of the health status of ADHF patients. This can lead to a better understanding of how ADMA influences the disease process and potentially guide treatment decisions or risk stratification. ”

Q11. Why did the authors choose to exclude patients with myocardial infarction less than 1 year prior index hospital admission?

R11. This exclusion criterion was implemented to ensure that the study focuses on the specific population of interest without confounding factors from recent myocardial infarctions. Patients with a recent MI, especially within the first year, often receive several medications aimed at preventing further cardiac events and managing associated risk factors. Such an example is Dual Antiplatelet Therapy (DAPT), combining aspirin with a P2Y12 inhibitor. This is often prescribed in our country for typically up to one year following an acute coronary syndrome or myocardial infarction, to further reduce the risk of recurrent ischemic events. While DAPT can be effective in preventing cardiovascular events, it's important to be aware of potential side effects, including their impact on renal, glycemic, lipid, and hepatic function.

Thus, DAPT, particularly aspirin, can potentially affect renal function, especially in patients with pre-existing kidney disease or those at risk of kidney injury. Aspirin can cause renal impairment by reducing renal blood flow, particularly in patients with decreased renal perfusion. P2Y12 inhibitors may also have renal effects, though they are generally considered to be less nephrotoxic than aspirin. On the other hand, DAPT, especially clopidogrel, has been associated with an increased risk of hyperglycemia and new-onset diabetes mellitus.

Q12. Was there any difference in the outcome between patients with high and low ADMA levels?

R12. Yes, based on the results of this study, there seems to be a difference in outcome between patients with high and low ADMA levels. More precisely, patients with high ADMA had a significantly higher chance of having ischemic cardiomyopathy, required a longer stay in the ICU compared, displayed a poorer kidney function, and less favorable lipid profile. These aspects have been already addressed in the initial version of the manuscript. It is, however, important to note that this is a single study and further research is needed to confirm these findings. For example, the study only looked at ADMA levels at admission, and it is possible that changes in ADMA over time could also be important.

Q13. Will authors suggest that ADMA can be used in the ICU as a prognostic biomarker?

R13. The present results suggest that ADMA might be useful for risk stratification in patients with heart failure admitted to the ICU . e.g., higher ADMA implies a higher likelihood for a longer hospital stay. Although tempting, however, (and based on the present knowledge on this topic) we cannot suggest that ADMA can be used at the present moment in the ICU as a prognostic biomarker. Nonetheless, ADMA's role would be to improve prognostication in combination with established methods, requiring further investigation.

Based also on the suggestions of the 1st reviewer, we have improved the final part of the Disscusion section from the first version of our manuscript by rewriting (cleaning the content) the last two pragraphs from this section (lines 403-434, first version of the manuscript) so that text was shortened. This action also aimed at avoiding conclusions/recommendations that are supported (only) by the findings of the present paper (lines 417-427, first version of the manuscript)

Q14. Methodology is well written, as well as discussion section and conclusion. results are clearly presented. References are adequate.

R14. Thank you very much for taking the time to review our manuscript. We are delighted to hear that you found the methodology, discussion section, and conclusion to be well-written. We invested significant effort into ensuring clarity and coherence in these sections, and we are pleased that our efforts have been recognized.

In addition, we have rewritten a large part of the initial manuscript (as suggested by the first two reviewers), improved grammar and English to improve the readability of text. We firmly believe that the Reviewer’s comments and suggestions have
significantly improved this manuscript. We hope that the Reviewer find the revised version of the manuscript suitable for publication in Medicina.

Round 2

Reviewer 1 Report

Comments and Suggestions for Authors

Authors substanstially improved their manuscript. I think there is still are for:

1) small improvements in English language (see below) 

2) definition of pulmonary hypertension due to left heart disease seems not appropriate as no invasive estimation of pulmonary vascular resistance was performed. 

3) It is not clear why the authors decided to report only baseline characteristics lacking significant differences among subgroups in Table 1 and I would again suggest to do it. 

Comments on the Quality of English Language

Abstract: please correct "subjects included 83 individuals"

Improvement in terminology use is still possibile throughout the text: e.g. Abstract line 30 contains redundant, yet not very specific, words "Selected cardiovascular, blood, glycemic, renal, lipidi, and hepatic health markers"

Author Response

Dear Reviewer,

Please find enclosed the second revised version of our manuscript; and  below our response letter explaining, point-by-point, the changes made in response to your suggestions . However, we, first sincerely thank  you for your thorough review of our manuscript and for the excellent suggestions that we received. We have made a concerted effort to appropriately respond to each of your suggestions.

Q1. small improvements in English language (see below)

R1. We have improved this part of the initial manuscript as suggested by the reviewer – please see the second revision of the manuscript.

Q2. definition of pulmonary hypertension due to left heart disease seems not appropriate as no invasive estimation of pulmonary vascular resistance was performed.

R2. In our country, as a general rule, the diagnosis of pulmonary hypertension is put via echocardiography using the criteria already mentioned in our previous response; that is, mean pulmonary artery pressure (mPAP) ≥ 25 mm, and/or systolic pulmonary artery pressure (sPAP) ≥ 35 mm Hg; and tricuspid regurgitant jet velocity > 3.4 m/s. Echocardiography is used as the first-line approach due to its safety, accessibility, and ability to provide sufficient information for diagnosis and management of pulmonary hypertension in these patients. If echocardiography findings are unclear or conflicting with clinical suspicion, the right heart catheterization — the gold standard for diagnosing pulmonary hypertension — is used; it is an invasive procedure, which carries a risk of complications like bleeding or infection; and is not readily available in healthcare settings from our county. The former drawback is also of significant concern for the high risk category of patients with chronic heart failure hospitalized due to recurrent decompensation episodes. That is why no invasive estimation of pulmonary vascular resistance was performed.

In addition, the definition of PH-LHD was changed to: ”PH-LHD was defined as known LHD or newly diagnosed LHD (as per national guidelines for LHD diagnosis/management) with a mean pulmonary artery pressure (mPAP) ≥ 25 mm, and/or systolic pulmonary artery pressure (sPAP) ≥ 35 mm Hg; and tricuspid regurgitant jet velocity > 3.4 m/s [35] ”

Q3. It is not clear why the authors decided to report only baseline characteristics lacking significant differences among subgroups in Table 1 and I would again suggest to do it.

R3. Thank you for your comment. We have chosen to report the results of inter-strata comparisons for categorical variables in Table 1; and those for continuous variables in Table 2 (and Figures 1 and 2). This approach — separate tables for categorical and continuous data  — aimed to improve clarity, readability, and effective communication of the results. We also mention that categorical variables showing significant inter-group differences are already included in Table 1 (see variable Ischemic cardiomyopathy in the first version of the revised manuscript).

For continuous variables, we have chosen to report the data for non-significant variables in Table 2; and those for significant variables in Figures 1, 2. This decision to separate significant and non-significant continuous variables in the results section reflects a common practice aimed at clarity and conciseness. Here's our reasoning:

 - Tables are well-suited for presenting non-significant comparisons. They provide a clear and concise overview of baseline characteristics for both groups, allowing readers to compare the distribution of values.

 - Significant findings often benefit from visual aids. Figures like scatter plots, boxplots, or bar charts can effectively highlight the magnitude and direction of the differences between groups for significant variables.

By separating the data, we aimed to: Improve readability: Readers can easily locate the key findings (significant differences) and explore the overall distribution of non-significant variables; Optimize space: Tables are more efficient for presenting numerous non-significant comparisons, while figures allow for a more impactful presentation of significant findings. In addition, for consistency in data presentation, we have already reported in text the results for the significant (continuous) variables (shown in Figures 1, 2) with all the details included in Table 2; that is median values and interquartile range.

We have inserted in the revised version of the manuscript a brief explanation why we have chosen to present significant and non-significant data separately.”By separately presenting data for non-significant (continuous) variables and significant (continuous) variables (see Table 1, and Figures 1 and 2), we aimed to improve readability, optimize space, and effectively highlight the magnitude and direction of inter-strata differences for significant variables. ”

Q4. Comments on the Quality of English Language Abstract: please correct "subjects included 83 individuals".

R4. Ok. We have made the appropriate changes in the second revision of the manuscript. the term ” subjects” was replaced with ”This pilot study”.

Thank you again very much for taking the time to review our manuscript. We are delighted to hear that you found the methodology, discussion section, and conclusion to be improved versus the first version of the revised manuscript. We invested significant effort into ensuring clarity and coherence in these sections, and we are pleased that our efforts have been recognized.

Anamaria Vîlcea

Department of Internal Medicine, Faculty of Medicine, ‘Vasile Goldiș’ Western University of Arad, Romania, Bulevardul Revoluției 94, Arad 310025, Romania.

Arad County Emergency Clinical Hospital, Romania, Str. Andrényi Károly Nr. 2‑4, 310037 Arad, Romania